# Competition between Winter Wheat and Cornflower (*Centaurea cyanus* L.) Resistant or Susceptible to Herbicides under Varying Environmental Conditions in Poland

Roman Wacławowicz [1,*], Ewa Tendziagolska [1], Agnieszka Synowiec [2], Jan Bocianowski [3], Cezary Podsiadło [4], Krzysztof Domaradzki [5], Katarzyna Marcinkowska [6], Ewa Kwiecińska-Poppe [7] and Mariusz Piekarczyk [8]

1 Institute of Agroecology and Plant Production, Wrocław University of Environmental and Life Sciences, pl. Grunwaldzki 24A, 50-363 Wroclaw, Poland
2 Department of Agroecology and Crop Production, The University of Agriculture in Kraków, Al. Mickiewicza 21, 31-120 Krakow, Poland
3 Department of Mathematical and Statistical Methods, Poznań University of Life Sciences, Wojska Polskiego 28, 60-637 Poznan, Poland
4 Department of Agroengineering, The West Pomeranian University of Technology in Szczecin, ul. Papieża Pawła VI 3, 71-459 Szczecin, Poland
5 Department of Weed Science and Soil Tillage Systems, Institute of Soil Sciences and Plant Cultivation—State Research Institute, ul. Orzechowa 61, 50-540 Wroclaw, Poland
6 Institute of Plant Protection—National Research Institute, ul. Władysława Węgorka 20, 60-318 Poznan, Poland
7 Department of Herbology and Plant Cultivation Techniques, University of Life Sciences in Lublin, Akademicka 13, 20-950 Lublin, Poland
8 Department of Agronomics, Faculty of Agriculture and Biotechnology, University of Science and Technology, Al. Kaliskiego 7, 85-796 Bydgoszcz, Poland
* Correspondence: roman.waclawowicz@upwr.edu.pl

**Abstract:** Competitive ability of cereals against segetal weeds depends among other things, on soil properties and the weather. Concerning cornflower (*Centaurea cyanus* L.), this issue is poorly recognized, as there are no reports on the impact of environmental conditions on the competitiveness of wheat against susceptible and resistant biotypes. Our study aimed to evaluate the effects of site and weather conditions on the competitive effects between winter wheat (WW) and two cornflower biotypes, either florasulam and tribenuron-methyl resistant (R) or sensitive (S). The experiment was conducted in a replacement series model at six sites across Poland in three growing seasons. The competitive relations were determined on the basis of two indices, i.e., the relative biomass and the number of seeds produced by the tested plants. The relative yield of wheat and weed were plotted on graphs and fitted to one of five competition models. In addition, a competitive ratio (CR) was calculated on the basis of fresh plant biomass and the number of seeds. Correlation coefficients were determined between the length of the plant, yield, biomass, the number of seeds per plant, hydrothermal coefficient K, and soil texture. Biometric parameters of wheat for its competition with two cornflower biotypes were analyzed using canonical variate analysis (CVA). The number of days to WW emergence and the day-difference between WW and cornflower (B) emergence were also calculated. The environmental characteristics of the sites, i.e., hydrothermal coefficient K and soil texture, were used as categorizing variables. Drought generally favored the greater competitive ability of WW against B for both biomass accumulation and seed production. During the first season of the research (relatively dry), only in one case out of 12 cases cornflower was more competitive than wheat. In the second year of the experiment (dry season), the competition of WW against B for resources was lower. It depended more on the site than on the cornflower biotype or the proportion of plants in the mixture. Under high or optimal rainfalls (the third year of the experiment), the competitiveness of WW toward B was significantly lower than in years with rainfall deficiency. In addition, the ability of wheat competition against the weed may have been influenced by the earlier emergence of wheat than cornflower. Even though it was sown together with wheat, cornflower emerged 0–12 days later than the tested cereal. It was also noticed that wheat was more competitive on light soils against the herbicide-susceptible (S) biotype. In contrast, greater WW competitiveness was observed against herbicide-resistant (R) cornflower on heavy soils. In conclusion, winter wheat

competitiveness against herbicide-resistant or herbicide-sensitive cornflower biotypes is significantly dependent on weather and soil conditions. It is therefore reasonable to study this phenomenon in more detail. It would also be interesting to learn more about the underground competition on varying soil types under different water availability.

**Keywords:** replacement series model; competitive ratio; soil type; weather conditions

## 1. Introduction

The occurence of weeds in crop fields is undesirable for different reasons. Weed infestation can, among other things, interfere with harvest, increase the crop's contamination and moisture, and even promote the spread of pests and pathogens [1]. However, the greatest damage is usually the result of competition between weeds and crops for nutrients, water, and light leading to a reduction in crop yield [2–7]. Crop-weed competition is also affected by factors such as the timing of weed emergence, weed density, and the type of weed species [8].

Many crop species cannot cope well with weed competition, and the advantage in gaining competition over weeds is significantly affected by environmental factors such as weather and soil conditions [9–11]. Moreover, weeds' physiological plasticity and conciderable advantage in interspecific genetic variability in comparison with most crops can give weeds a competitive advantage under changing environmental conditions, such as floods, droughts, or extreme temperatures [12]. Weeds grow rapidly and use all available environmental resources more efficiently than crops. In addition, in simplified crop rotations of increased agrochemical inputs, herbicide-resistant weeds can appear in the canopy, threatening crop yields and biodiversity [13] by their dominance [14,15]. In other research, the authors [16] noted that plant resistance to water stress is linked to the expression of LACS genes (long-chain acyl-CoA synthetases), which play an important role in water loss control. These genes, contributing to the biosynthesis of cutin and suberin, are potential targets for genetically modifying the qualities of crops. In Poland, cornflower (*Centaurea cyanus* L.), which belongs to the Asteraceae family, originated in the Caucasus [17], is one of the herbicide-resistant weeds. This species has developed resistance to herbicides from the group of acetolactate synthase (ALS) inhibitors (HRAC group 2) and synthetic auxins (2,4-D and dicamba) (HRAC group 4) [18–20]. Additionally, tribenuron methyl and florasulam-resistant cornflower has recently been found [19,21].

Moss et al. [22] created a weed species catalog with a high or medium risk of resistance evolution. *Centaurea cyanus* L. is not on the list, so it is perceived as a low-risk species. Moreover, cornflower is classified as a minor weed or even an endangered species in Western Europe due to agricultural intensification in the last decades [23–27]. However, such species will not always behave equally in all areas of the world, and the possibility of acquiring resistance could vary over time [22]. Thus, a weed species with a low risk of acquiring resistance may become an issue. In Poland *Centaurea cyanus* L. is an example of species which is common throughout the country. Although the cornflower prefers light soils, it appears in increasing numbers also on heavier soils, infesting winter wheat and oilseed rape [28]. The species is also noted in fallows, wasteland, and ruderal sites. It is an invasive species in grassland prairies in the US, along railroads and field margins [29,30]. The economic threshold for cornflower in cereals is equal to 1–5 plants m$^{-2}$. In Poland, cornflower is becoming an increasing problem due to the popularity of growing winter wheat in succession with winter oilseed rape and the small range of herbicides that effectively eliminate this weed [21]. One cornflower plant in a winter crop can produce more than a thousand achenes, which fall off quickly, usually before harvest, and retain their germination capacity for 5–10 years [17,31]. The achenes germinate in the fall, at the same time as the winter crop, and hibernate in rosette form [32]. What is interesting is that cornflower, apart from being a problem as a weed, is also an ornamental plant. Its blue petals have

been applied in food preparation, as a decorative and potential source of natural coloring purposes. [33–35]. Also, this species has been used in medicine and herbalism for centuries as an diuretic and anti-inflammatory (particularly in ophthalmology) agent [25,34].

The competitiveness of wheat against cornflower is poorly recognized, especially since there is no reports concerning on the impact of environmental conditions on that issue toward herbicide-susceptible or resistant cornflower biotypes. Such studies are important as they can predict yield losses due to the presence of a weed [36]. Many methods are known for studying competitiveness between different plant species [37–40]. Of the four most commonly presented models (additive, substitutive (replacement), systematic, and neighborhood) proposed by Radosevich [36], the most suitable for our experiment is the replacement series model [41], where the total plant density is kept constant. At once, the mixture proportions for the wheat-cornflower mixture are varied. The model makes it possible to assess which species or biotypes are more competitive with others [36,42]. Based on a replacement series model, herbicide-resistant *Apera spica-venti* L. was more competitive with wheat than the susceptible biotype [9]. In other studies, wheat generally suppressed herbicide-resistant and herbicide-susceptible *Alopecurus myosuroides* Huds. However, the authors showed that, on average, the susceptible (S) biotype was more competitive against the resistant (R) biotype [43].

In the available literature, research on competition between herbicide-resistant and susceptible cornflowers against winter wheat is missing. Therefore, a study was undertaken to evaluate the outcome of environmental conditions (weather and soil) on competitive effects between winter wheat and two cornflower biotypes (S and R).

## 2. Materials and Methods

In three-year pot experiments conducted at six different regions of Poland (Figure 1), two biotypes (B) of cornflower (*Centaurea cyanus* L.) with opposing herbicide sensitivity were tested: susceptible (S) and herbicide-resistant (R) (Table 1). The competitive effects of the cornflower biotypes were examined against winter wheat (WW) cv. Arkadia (breeder: HR Danko PL).

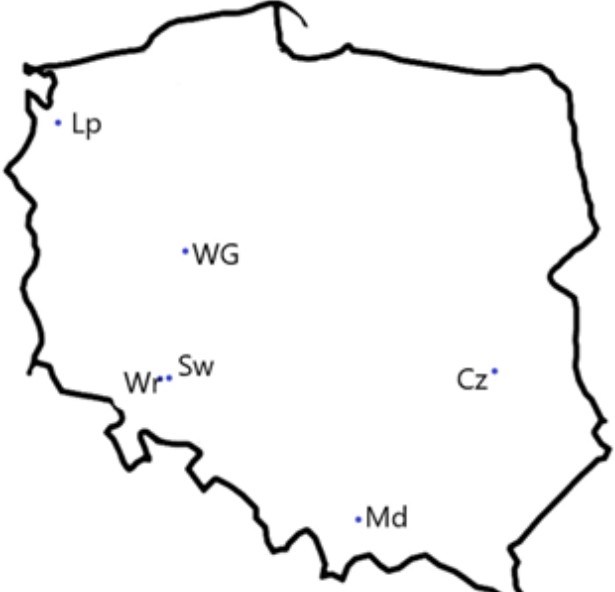

**Figure 1.** Study site distribution. Lp—Lipnik; WG—Winna Góra; Cz—Czesławice; Sw—Swojczyce; Wr—Wrocław; Md—Mydlniki.

**Table 1.** Characteristics of herbicide-resistant (R) and -susceptible (S) cornflower biotypes (*Centaurea cyanus* L.) used in the pot experiments.

| Biotype | Herbicide (HRAC Group) | |
| --- | --- | --- |
| | Florasulam (HRAC 2) | Tribenuron Metylu (HRAC 2) |
| R | RRR (123 *) | RRR (>480) |
| S | S (0.865) | S (1.85) |

* The numbers in brackets relate to the active ingredient's effective dose (g ha$^{-1}$), causing a 50% reduction in plant biomass (ED50); S—susceptible; RRR—highly resistant.

According to the replacement series competition model [9,43,44], the pot experiments were designed and conducted across Poland in situ at six different sites with varying physical and chemical soil properties in three replications everywhere and took place in the next three growing seasons. The first experiment was conducted in the 2017–2018 season at three different sites (Winna Góra, Swojczyce, and Wrocław), the second one during the 2018–2019 season at five sites (Lipnik, Czesławice, Swojczyce, Wrocław, and Mydlniki), and the third one (2019–2020) at two sites (Czesławice and Mydlniki) (Table 2).

**Table 2.** Coordinates and soil physical and chemical properties of the study sites.

| Site | Latitude | Longitude | Particles (%) | | | Texture | N [1] | P | K | OM | pH |
| --- | --- | --- | --- | --- | --- | --- | --- | --- | --- | --- | --- |
| | | | Sand | Silt | Clay | | | | | | |
| Lipnik | 53°34′ N | 14°95′ E | 85.4 | 14.0 | 0.6 | Loamy sand | 0.11 | 232 | 301 | 2.2 | 6.4 |
| Winna Góra | 52°12′ N | 17°26′ E | 70.4 | 26.3 | 3.4 | Sandy loam | – | 137 | 162 | 1.1 | 5.5 |
| Czesławice | 51°18′ N | 22°16′ E | 15.8 | 72.6 | 11.6 | Silt loam | 0.12 | 158 | 196 | 1.5 | 6.4 |
| Swojczyce | 51°06′ N | 17°08′ E | 66.0 | 26.0 | 8.0 | Sandy loam | 0.51 | 128 | 125 | 1.2 | 6.5 |
| Wrocław | 51°04′ N | 17°02′ E | 56.0 | 23.0 | 21.0 | Sandy clay loam | 0.70 | 182 | 197 | 1.2 | 6.2 |
| Mydlniki | 50°07′ N | 19°84′ E | 39.0 | 52.0 | 9.0 | Silt loam | 0.07 | 173 | 196 | 1.1 | 6.3 |

[1] N nitrogen (%), P phosphorus (mg kg$^{-1}$), K potassium (mg kg$^{-1}$), OM organic matter (%).

The pots scheme and the plant distribution per pot were the same each year and at each site. At each site, 36 pots (diameter 22 cm, 7 L vol., 0.038 m$^2$) were dug into the soil so that ca. 2.5 cm was above the soil surface. The pots were located at a 0.75 m distance. Next, after the soil was taken from the arable fields from layer 0-30 cm, it was sieved and places in pots. The content of basic macronutrients, pH, OM in the soil were determined as well as the soil texture (Table 2). The experiment was bird-proofed by attaching dense-mesh plastic net. In addition black plastic sheeting was laid between the pots to protect the area against weeds.

The seeds of winter wheat (WW), as well as herbicide-resistant cornflower (BR) and herbicide-susceptible cornflower (BS), were sown in one day, which was optimal for WW. The proportions of WW and BR were in the ratios of WW10:R0, WW8:R2, WW6:R4, WW4:R6, WW2:R8, and WW0:R10. The same ratios were used for the WW and BS plants: WW10:S0, WW8:S2, WW6:S4, WW4:S6, WW2:S8, and WW0:S10 (Table 3). Several WW seeds were sown per pot at a depth of 2 cm and cornflowers (S and R) at a depth of 1.0 cm. Each spot of the sowing plant was marked.

**Table 3.** Hydrothermal coefficient (K) calculated for the study sites for the seasons 2017/18, 2018/19 and 2019/20.

| Study Site | 2017/18 | Classification | 2018/19 | Classification | 2019/20 | Classification |
|---|---|---|---|---|---|---|
| Lipnik | - | - | 0.5 | Dry | - | - |
| Winna Góra | 1.2 | Relatively dry | - | - | - | - |
| Czesławice | - | - | 1.0 | Dry | 2.0 | Humid |
| Swojczyce | 1.2 | Relatively dry | 1.1 | Relatively dry | - | - |
| Wrocław | 1.0 | Dry | 0.7 | Dry | - | - |
| Mydlniki | - | - | 1.8 | Relatively humid | 1.5 | Optimal |

In the spring, according to the arrangement of the experiment, the density of wheat and cornflower was regulated. N fertilization was applied in two rates: 15.0 g $NH_4NO_3$ $m^{-2}$ at the beginning of the growing season, and the same rate was applied at wheat shooting (BBCH 31–33).

Wheat was harvested at full grain maturity (BBCH 89), and cornflower was harvested at achenes ripening (BBCH 80-85). Then the fresh weight of both species was weighed. For wheat, the number and the weight of grains as well as the weight of 1000 grains (TGW) were calculated. As far as cornflower is concerned, the number of seeds per plant was also calculated. For this purpose, anthodiums were counted per plant. Based on 10 anthodiums from each plant, the average number of achenes (seeds) per anthodium was determined. The proportion was calculated as the average number of seeds per cornflower (B).

*2.1. Weather Conditions*

Weather data, i.e., air temperature and precipitation, were collected during the study from the nearest weather stations (Tables S1–S3). The weather data for the months from WW sowing until harvesting, with temperature >0 °C, namely October–December 2017 and April–July 2018, October–November 2018 and April–July 2019, as well as October–November 2019 and April–July 2020, were valorized using the hydrothermal coefficient (K) according to the equation:

$$K = 10P/t \qquad (1)$$

where $P$ is the precipitation total, and $t$ is the sum of the daily mean air temperature values. The classification for Poland's temperate climate is $K \leq 0.4$ extremely dry; $0.4 < K \leq 0.7$ very dry; $0.7 < K \leq 1.0$ dry; $1.0 < K \leq 1.3$ quite dry; $1.3 < K \leq 1.6$ optimum; $1.6 < K \leq 2.0$ quite humid; $2.0 < K \leq 2.5$ humid; $2.5 < K \leq 3.0$ very humid and >3.0 extremely humid [45]. The calculated K values are presented in Table 3.

*2.2. Statistical Analysis*

The number of seeds per plant and fresh plant biomass were calculated for winter wheat (WW) and cornflower (B), independently. The analysis was carried out for the replacement series experiment [42,46]. The relative yield (RY) of WW ($RY_{WW}$) was calculated according to the following formula:

$$RY_{WW} = (p)\,(WW_{mix}/WW_{mon}) \qquad (2)$$

where $p$—the proportion of species, $WW_{mix}$—the value of the WW parameter analyzed for the mixture, $WW_{mon}$—the value of the W parameter analyzed for the monoculture. The relative yield (RY) of B ($RY_B$) was calculated according to the formula:

$$RY_B = (p-1)\,(B_{mix}/B_{mon}) \qquad (3)$$

where $B_{mix}$—the value of the B parameter analyzed for the mixture, $B_{mon}$—the value of the B parameter analyzed for the monoculture. RY values for WW and B show the mean value for an individual plant in the pot.

The total relative yield (TRY) was calculated according to the following formula:

$$TRY = RY_{WW} + RY_B \tag{4}$$

Two parameters: fresh plant biomass and the number of seeds per plant, were measured for both WW and B. According to Radosevich [36], these parameters were calculated into RY and TRY, presented as graphs, and fitted into one of the five competition models. The comparisons between empirical and theoretical competition models for the plants' biomass and the number of seeds per plant were tested by *t*-test independently for each biotype, year, and site [36,47]. Moreover, the competitive ratio (CR), representing the comparative growth of WW compared to B, was calculated based on fresh plant biomass (CRb) and the number of seeds (CRse), according to the following formula:

$$CR = ((1-p)/p)/(RY_{WW}/RY_B) \tag{5}$$

where RY is the relative yield, and p is the proportion of biotype. If CR >1, it means WW was more competitive toward B.

Shapiro-Wilk's normality test was used to testing of the normality of distribution of CRb and CRse [48]. Bartlett test was used to testing of the homogeneity of variance. Homogeneity of variance-covariance matrices and multivariate normality were tested using the Box's M test. Three-way analysis of variance (ANOVA) were carried out to determined the main effects of site, the proportion of plants in the mixture, and biotype as well as their interactions on the variability of the particular traits, independently for 2017–2018, 2018–2019, and 2019–2020. The relationships between the CRb and CRse were estimated based on Pearson's correlation coefficients and tested for all sites separately in each tested year.

Relationships between length, yield, TGW, biomass, seeds, K, and sand were assessed using Pearson's correlations separately for 2017–2018, 2018–2019, and 2019–2020.

Additional biometric parameters of WW in competition with RB and RS in 2017–2018, 2018–2019, and 2019–2020 were analyzed separately using the canonical variate analysis (CVA) [49,50]. Discriminant analysis was carried out to determine the relative share of each original trait in the multivariate variation of the treatments using the Pearson's correlation coefficients. Grain yield and biomass of wheat, the number of grains per plant, WW length and TGW were taken into account in the calculations. These parameters were expressed as relative value (RY). The time of wheat emergence and the number of days between wheat and cornflower emergence were also the components of the analysis. The categorization variables were parameters defining weather (K-factor) and soil conditions (texture).

The GenStat v. 18.2 (Hemel Hempstead, UK) statistical software package was used for all the analyses.

## 3. Results

Winter wheat emergence (WW) was observed 8–12 days after sowing in 2017, 8–15 days in 2018, and 15–18 days in 2019 (Table 4). In 2017 wheat emerged the fastest in Swojczyce (after eight days). At this site in October, precipitations (71.4 mm) and air temperature (11.2 °C) was the highest among other sites (Table S1). In contrast, this growing season recorded the longest emergence time (12 days) in Winna Góra, where precipitations were the lowest (44.1 mm).

In 2018 the fastest emergence (after eight days) was noticed in Lipnik and Mydlniki (Table S2). In October, around Szczecin (north of Poland), the weather was relatively dry and warm, while it was wet and cold in the Kraków (south of Poland) area. In 2018, wheat took the longest to emerge in Czesławice (15 days), where the lowest average air temperature (9.2 °C) was recorded. Extended wheat emergence was also noted in 2019. In Czesławice, wheat emerged after 15, and Mydlniki 18 days, most likely due to low autumn rainfall (37–38 mm) (Table S3). Although the sowing date for both wheat (WW) and cornflower (B) was the same, in 2017, B emerged 3–11 days later than WW (Table 4).

In 2018, cornflowers germinated in three sites 4–7 days after wheat emergence. In the last year of the study (2019), WW and B emerged in Czesławice and Mydlniki on the same day.

**Table 4.** Dates of winter wheat and cornflower emergence, and the number of days from wheat sowing until emergence.

| Site | 2017 | | | | 2018 | | | | 2019 | | | |
|------|------|------|------|------|------|------|------|------|------|------|------|------|
| | WW | DE | B | EB | WW | DE | B | EB | WW | DE | B | EB |
| Lipnik | - | - | - | - | 25.10 | 8 | 25.10 | 0 | - | - | - | - |
| Winna Góra | 30.10 | 12 | 05.11 | +6 | - | - | - | - | - | - | - | - |
| Czesławice | - | - | - | - | 22.10 | 15 | 22.10 | 0 | 24.10 | 15 | 24.10 | 0 |
| Swojczyce | 26.10 | 8 | 06.11 | +11 | 23.10 | 10 | 30.10 | +7 | - | - | - | - |
| Wrocław | 07.11 | 11 | 10.11 | +3 | 14.10 | 11 | 18.10 | +4 | - | - | - | - |
| Mydlniki | - | - | - | - | 09.10 | 8 | 13.10 | +4 | 24.10 | 18 | 24.10 | 0 |

WW—winter wheat; DE—days to WW emergence; B—cornflower; EB—the emergence of B to WW (average, days).

In the first season, wheat competition against the biomass of cornflower (R) or susceptible biotype (S) was examined (Figure 2, Table S4). The results show that in Winna Góra (for R and S) as well as in Swojczyce (for S), the competitive model III occurred, representing the mutual antagonism of the species being evaluated), pointing out two-sided negative significant impacts of competition on WW and R or S biomass. On the other hand, in Wrocław (for R and S) and Swojczyce (for S), a model I was recorded, which indicates the lack of competition between cornflower (B) and wheat (WW) in terms of biomass accumulation. It is also possible to interpret that the ability of one species to disrupt another is equivalent.

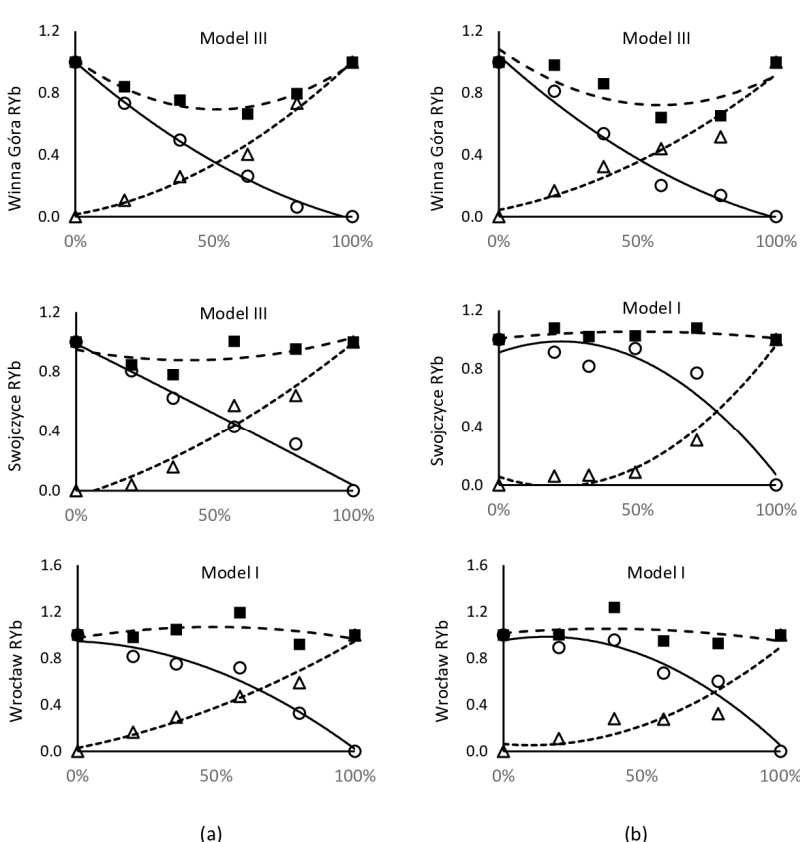

**Figure 2.** The replacement competition model for the relative biomass yield (RYb) of winter wheat and cornflower (B) in the 2017–2018 season (classified according to the *t*-test and *p* < 0.05). (**a**) competition between WW and B resistant (R) and (**b**) competition between WW and herbicide-susceptible cornflower (S). Legend: o—WW; Δ—R (**a**) or S (**b**), □—WW + R (**a**) or WW + S (**b**).

The competition between wheat and both cornflower biotypes, as demonstrated by the number of seeds per plant (Figure 3, Table S5), revealed distinct patterns than for plant biomass. In Swojczyce and Wrocław, model IIb was recorded for both cornflower biotypes (R and S). WW was more competitive than cornflower (B); the higher relative productivity of WW was indicated in the mixture (convex line) than the relative yield of B (concave line). In Winna Góra, on the other hand, there was no competition in seed production between each of tested cornflower (R or S) and winter wheat (model I).

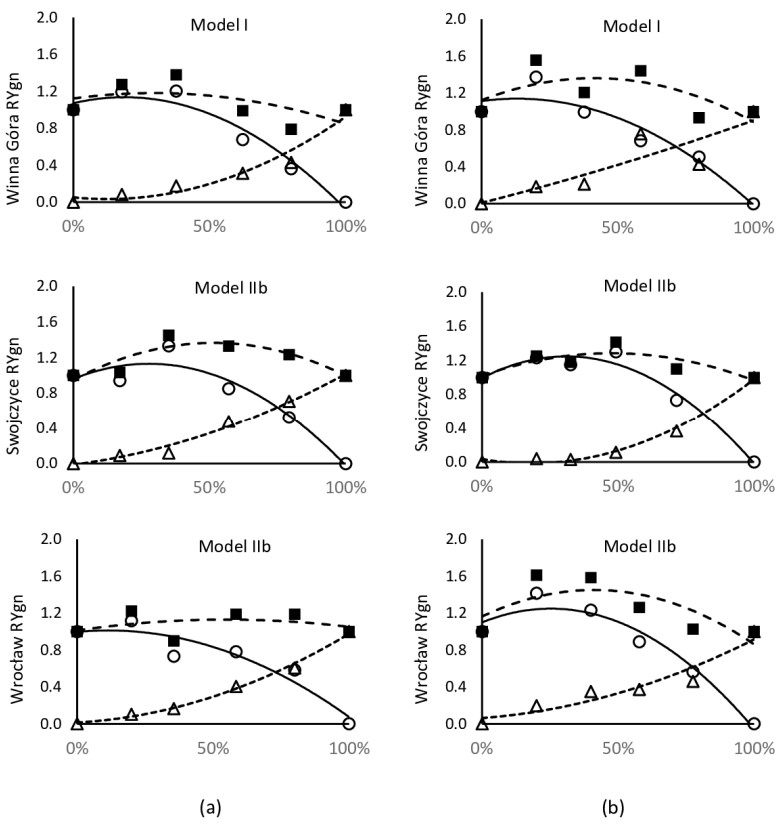

(a)   (b)

**Figure 3.** The replacement competition model for the relative seed number (RYse) of winter wheat and cornflower (B) in the 2017–2018 season (classified according to the *t*-test and *p* < 0.05). (**a**) competition between WW and B resistant (R) and (**b**) competition between WW and herbicide-susceptible cornflower (S). Legend: o—WW; Δ—R (**a**) or S (**b**), □—WW + R (**a**) or WW + S (**b**).

The competitive ratio (CR) in the plant ratios of 6WW:4B and 4WW:6B was measured for two parameters, i.e., biomass and seed number of WW and B. The ANOVA indicated that site and site × biotype interaction were statistically significant for both CRs. Additionally, biotype was significant for biomass (Table S6).

The CRse correlated with the CRb in the 2017–2018 season (*r* = 0.615, *p* = 0.033) (Figure 4). The WW's competitive effects were higher in the grain number (CRse) than in the biomass (CRb). In the first growing season, only cornflower (biotype S) was more competitive than wheat (CRse < 1; CRb < 1) at the Winna Góra site, provided that the proportion of WW in the pot was 0.4. Interestingly, in this locality, wheat showed greater competitive effects in seed number if it was in a mixture with the R biotype. But in Wrocław, the site effect predominated the cornflower biotype and the share of WW in the mixture. This became apparent in the graph in the form of a prominent cluster—the competitiveness indices calculated for the relative number of seeds (CRse) were within a narrow range of 2.3–3.2 regardless of the tested combinations. In Swojczyce, the competitive effects of wheat were greater in both seed number (CRse) and biomass (CRb), if WW was present in a mixture with the cornflower S biotype, regardless of the wheat proportion.

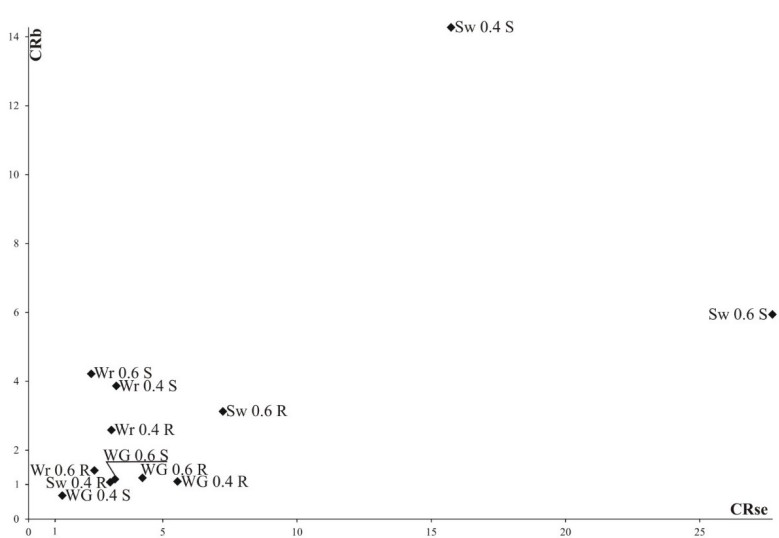

**Figure 4.** Relationships between relative biomass of plants (CRb) and the relative number of seeds (CRse) for the combinations of sites × winter wheat proportion in the mixture × cornflower biotypes, in 2017–2018. Legend: Sw—Swojczyce, WG—Winna Góra, Wr—Wrocław (sites of experiments); R—cornflower resistant biotype; S—cornflower susceptible biotype.

The correlation coefficients were presented as a heatmap for all pairs of observed features (length, yield, TGW, biomass, hydrothermal coefficient, and soil texture). Positive correlations were observed between followed pairs of traits: length-TGW ($r = 0.652$). However, negative for: length-K ($r = -0.636$), length-sand ($r = -0.614$), TGW-K ($r = -0.889$), TGW-sand ($r = -0.872$), biomass-K ($r = -0.578$) and biomass-sand ($r = -0.609$) (Figure 5).

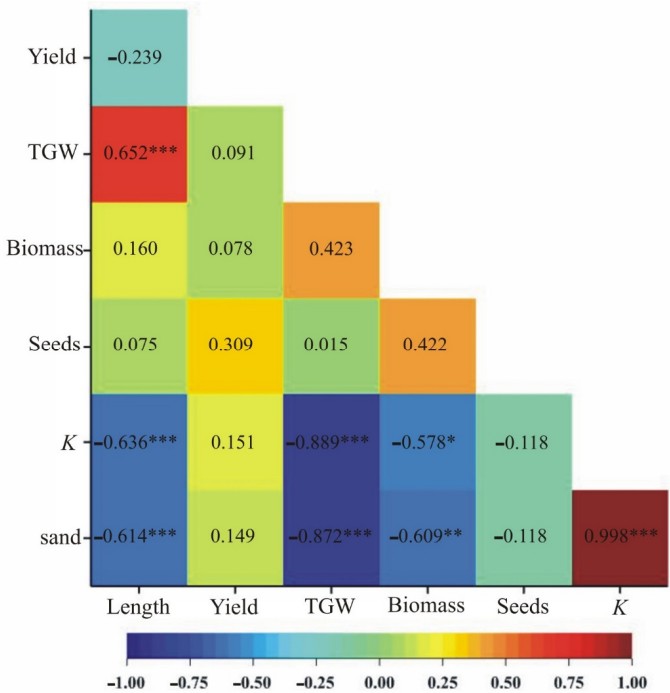

**Figure 5.** Heatmaps matrix for relationships between observed traits constructed on the basis of linear Pearson's correlation coefficients, in season 2017–2018. * $p < 0.05$; ** $p < 0.01$; *** $p < 0.001$.

The similarities in R and S competitive effects toward WW between different study sites in 2017–2018 was made on the basis of CVA. The first two ($V_1$ and $V_2$) canonical variables accounted for 96.64% of the total variability between the combinations of studied factors (Figure 6). The $V_1$ was significantly positively correlated with yield and TGW. The

$V_2$ was negatively correlated with the length of the plant, TGW, and the number of grains per plant (Table S7).

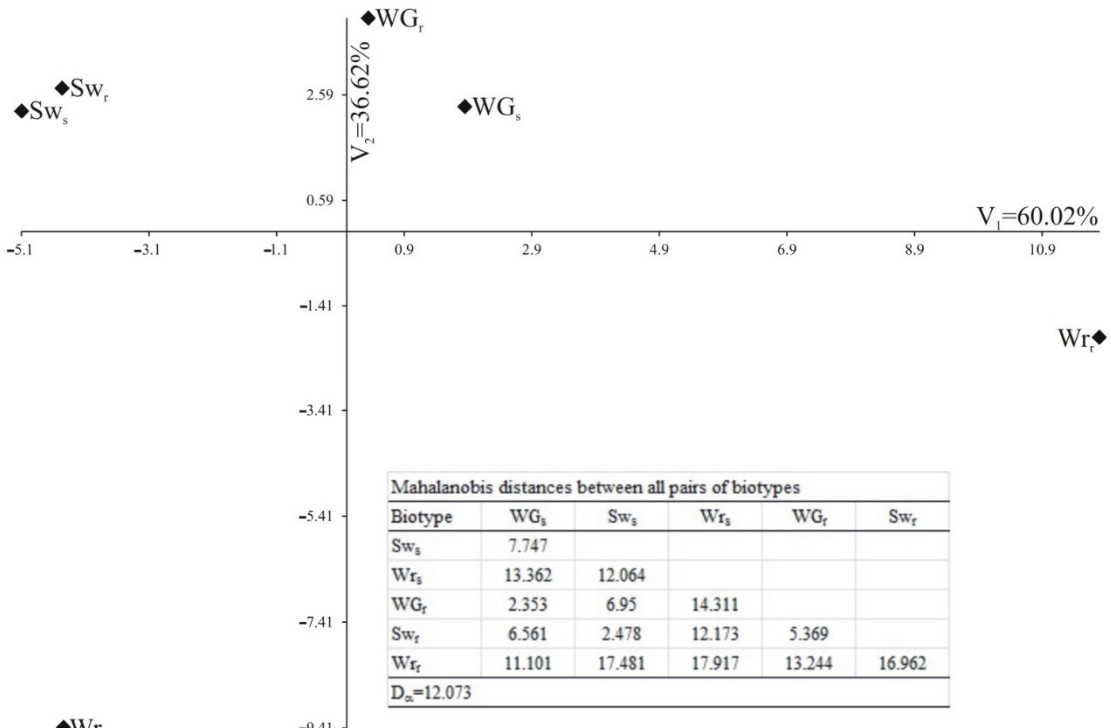

**Figure 6.** Distribution of combinations of sites and herbicide-resistance (R)/susceptibility (S), based on parameters of winter wheat (WW), in the space of the first two canonical variables ($V_1$ and $V_2$), in 2017–2018. Sites: Sw—Swojczyce, WG—Winna Góra, Wr—Wrocław. Subscript "r" denotes the competition of WW with R and the subscript "s" with S.

In the 2018–2019 season, the competition between the tested plant species varied depending on the site and the cornflower biotype (Figure 7, Table S8). A typical competition model (IIb) between winter wheat (WW) and cornflower (B) appeared in Lipniki (for biotype R) and Czesławice (for biotype S). In these sites, wheat was more competitive in forming a higher relative biomass yield than cornflower. In Mydlniki, the competition between wheat (WW) and the herbicide-susceptible cornflower (S) was described by model III. That model represents the mutual antagonism of the evaluated plants. At this site, none of the tested plant species made the expected contribution to biomass yield formation. In Lipniki, however, for wheat and the S cornflower, mutual benefits in competition between species in relative biomass yield were proven (model IV). Both species in the mixture produced relatively more biomass than would be expected in pure stands. In other sites, there was no competition in biomass production between R or S and wheat (model I).

For the herbicide-susceptible cornflower biotype (S), in as many as four out of five sites, the competitive relationship between WW and B expressed by the number of seeds in the plant (Figure 8, Table S9) showed the same relationship as for biomass. A different competition model was noticed only in Lipniki (model IIb), where wheat (WW) was dominant in forming the number of seeds. On the other hand, for the herbicide-resistant cornflower biotype, the concurrent model of competition between WW and B for the number of seeds and previously shown biomass occurred in three out of five sites. A different relationship was observed in Wrocław. Under these conditions, mutual benefits in competitive relationships for WW and B were noted in the formation of relative seed numbers (model IV). In Mydlniki, on the other hand, antagonism between WW and B was found, resulting in mutual losses for the species (model III).

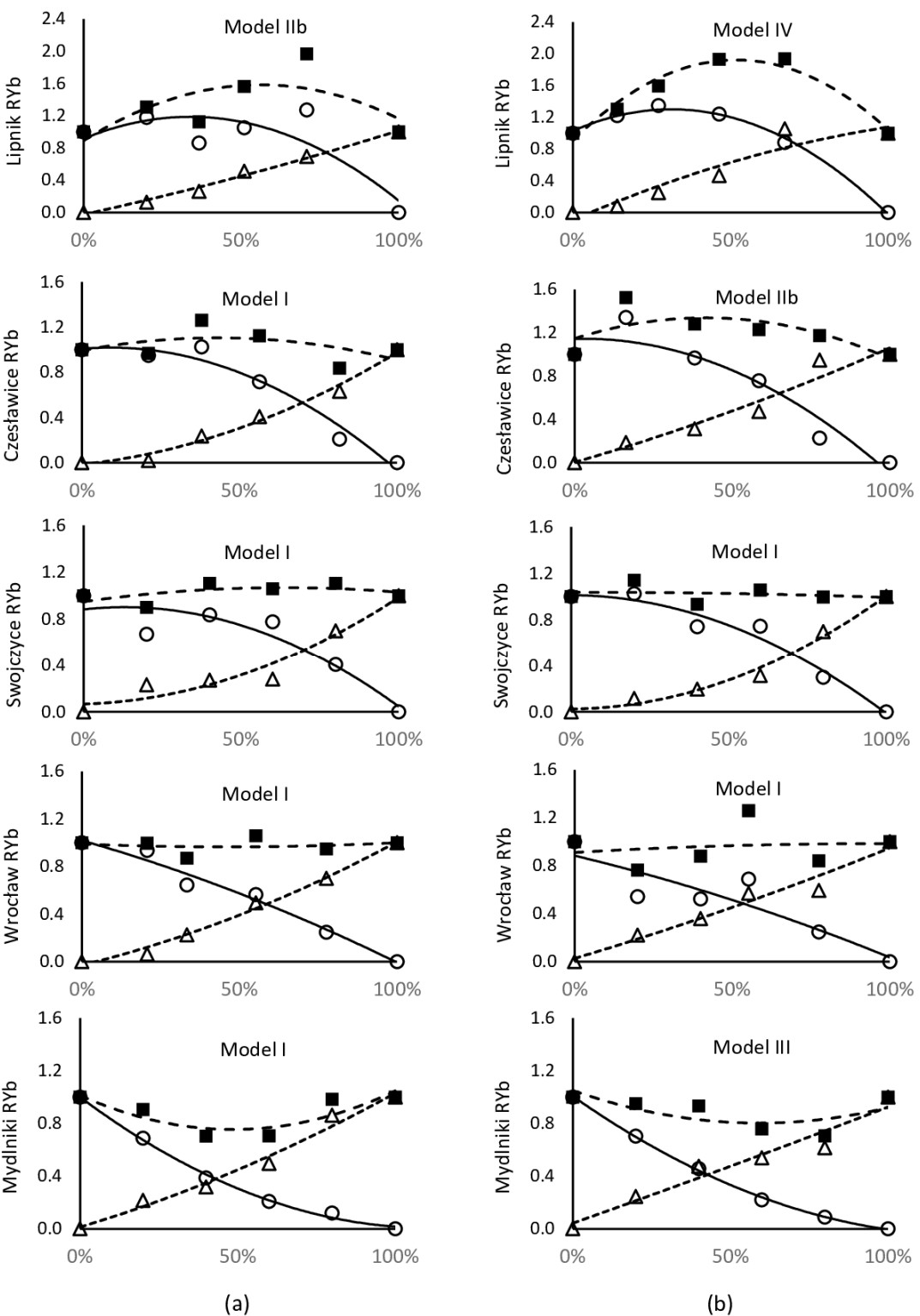

**Figure 7.** The replacement competition model for the relative biomass yield (RYb) of winter wheat and cornflower (B) in the 2018–2019 season (classified according to the *t*-test and $p < 0.05$). (**a**) competition between WW and B resistant (R) and (**b**) competition between WW and herbicide-susceptible cornflower (S). Legend: o—WW; Δ—R (**a**) or S (**b**), □—WW + R (**a**) or WW + S (**b**).

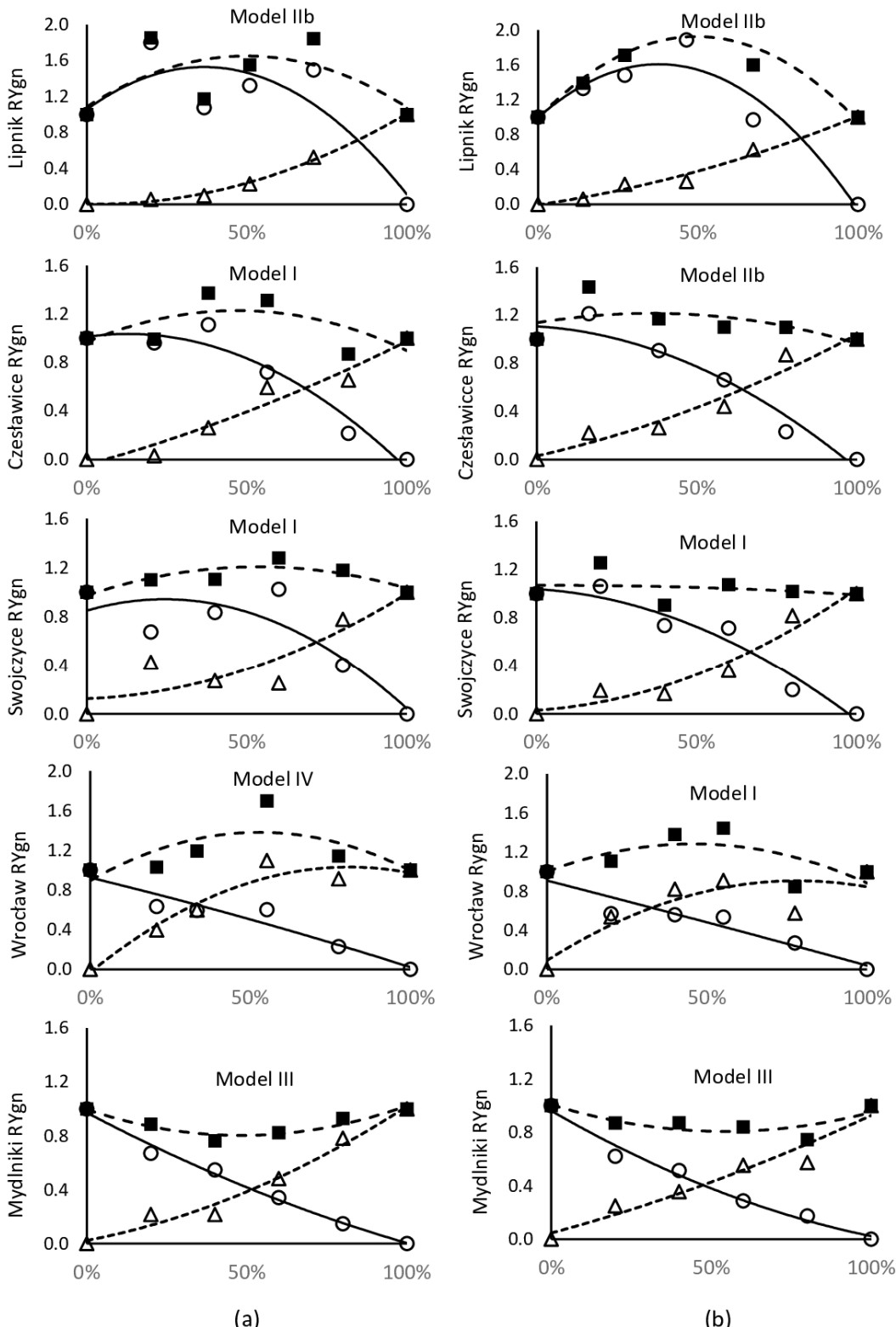

**Figure 8.** The replacement competition model for the relative seed number (RYse) of winter wheat and cornflower (B) in the 2018–2019 season (classified according to the *t*-test and *p* < 0.05). (**a**) competition between WW and B resistant (R) and (**b**) competition between WW and herbicide-susceptible cornflower (S). Legend: o—WW; Δ—R (**a**) or S (**b**), □—WW + R (**a**) or WW + S (**b**).

ANOVA indicated that the site and biotype effects, as well as site × biotype interaction, were statistically significant for CRb competitive index. Additionally, the main effect of site was significant for biomass (CRb) (Table S10).

Also, in the second year of the study (2018–2019), CRs were significantly correlated with CRb ($r = 0.721$, $p < 0.001$) (Figure 9). The analysis showed that the competition of wheat (WW) against cornflower (B) was higher in seed numbers (CRse) than in biomass (CRb). Interestingly, the competitiveness of wheat against cornflower was generally lower than in the previous season. A competitive index value of <1 for the number of seeds that occurred in Wrocław (for both B biotypes and WW shares in mixtures). Also, in Mydlniki, cornflower was more competitive with WW (CRse < 1), but only in the S biotype and WW share of 0.4. Concerning biomass, there was a significant impact of the site on interspecies competitiveness. The highest competitive effects of cornflower against wheat were observed in Mydlniki (CRb ranging from 0.7 to 1.0) and slightly higher in Wrocław (CRb from 0.9 to 1.5). Average values were recorded in Czesławice (1.8–3.5) and Lipniki (2.0–3.5), and the highest in Swojczyce (3.4–6.0). In Mydlniki, the S biotype cornflower found in both mixtures was more competitive with WW than the R biotype. This relation was observed especially concerning the number of seeds. In Lipniki, on the other hand, the R biotype has shown greater competitive effects than the S biotype. However, this relationship was observed only against biomass (CRb) and if the proportion of wheat in the mixture was 0.6. The opposite relationship was observed for the number of seeds (CRse).

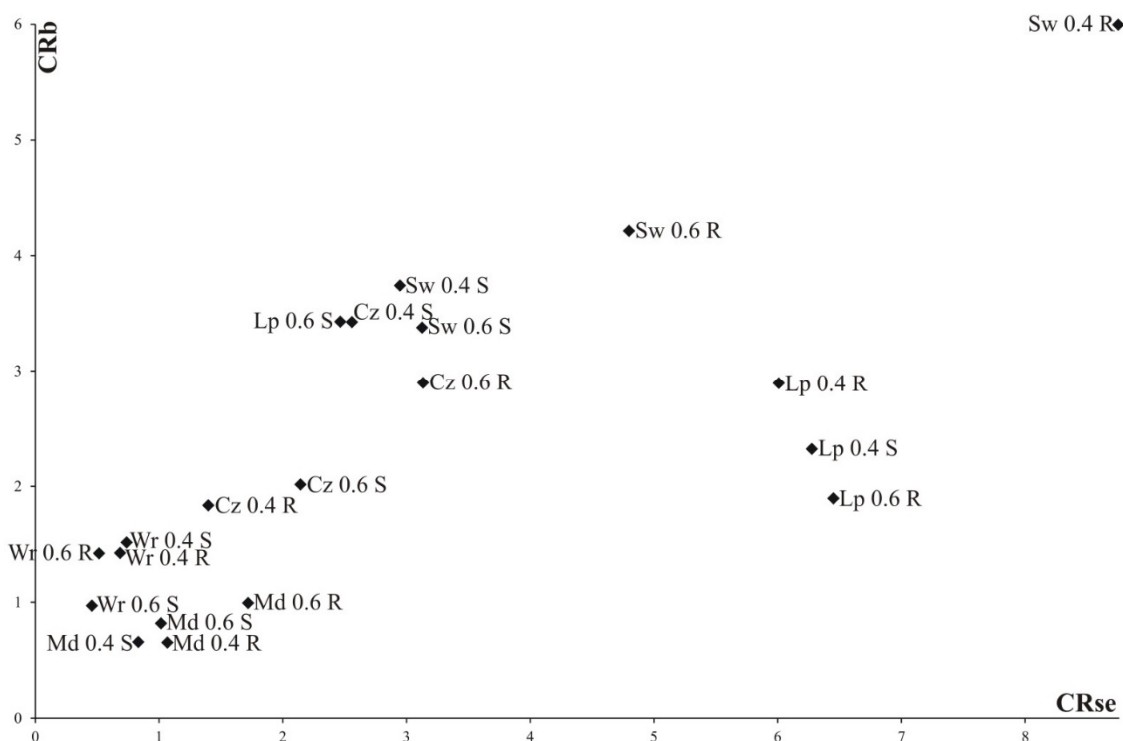

**Figure 9.** Relationships between relative biomass of plants (CRb) and the relative number of seeds (CRse) for the combinations of sites × winter wheat proportion in the mixture × cornflower biotypes, in 2018–2019. Legend: Cz—Czesławice, Lp—Lipniki, Md—Mydlniki, Sw—Swojczyce, WG—Winna Góra, Wr—Wrocław (sites of experiments); R—cornflower biotype of resistant cornflower; S—susceptible biotype.

In the second growing season (2018/2019), the relationships between mean values for trees of the observed features were also presented as a heatmap (Figure 10). Positively correlations were observed between followed pairs of traits: biomass-yield ($r = 0.928$), seeds-yield ($r = 0.958$), sand-yield ($r = 0.556$), biomass-seeds ($r = 0.919$), biomass-sand

($r = 0.524$), seeds-sand ($r = 0.628$). However, negative for: length-yield ($r = -0.389$), yield-K ($r = -0.602$), biomass-K ($r = -0.721$) and seeds-K ($r = -0.659$).

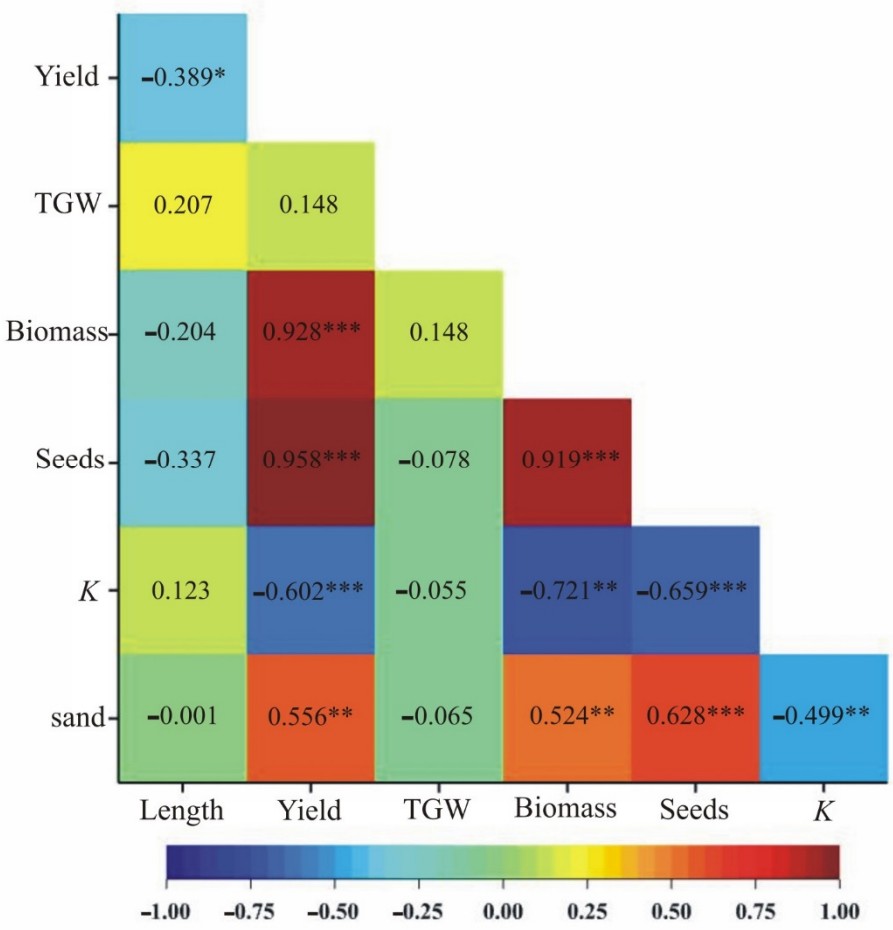

**Figure 10.** Heatmaps matrix for relationships between observed traits constructed on the basis of linear Pearson's correlation coefficients, in season 2018–2019. *, **, ***— significant at level 0.05, 0.01, 0.001, respectively.

The $V_1$ and $V_2$ accounted for 95.90% of the total multi-variability between the individual combinations (Figure 11). Positively correlation with $V_1$ was observed for: yield, plant biomass, and the number of grains per plant, however with $V_2$ for TGW (Table S11). In the second year (2018–2019), WW came out the best in competition with biotype S in Lipniki (dry season and loamy sand soil) as well as with biotype R in Czesławice (dry season and silt loam soil). In contrast, the weakest competitive capabilities of WW were found in Mydlniki (relatively humid season and silt loam soil) in competition with both biotypes (R and S), and in Wrocław (dry season and sandy clay loam soil) with biotype S. The greatest variation of all the seven traits jointly measured with Mahalanobis distances was found for $Lp_r$ and $Md_s$ (23.176). The greatest similarity was found between Mdr and Mds (2.159) (Figure 10).

In the third growing season (2019–2020), competition between cornflower (B) and wheat (WW) in the formation of relative biomass yield depended on both site and cornflower biotype (Figure 12, Table S12). In Mydlniki, a model I was shown for biotype R, indicating a lack of competition between cornflower and WW in biomass accumulation. In contrast, model III was determined for biotype S at the same site, representing mutual antagonism between the studied plant species in biomass yield formation. In Czesławice, the competitive relationship was the same for both biotypes (R and S). They are described by model IV, which indicates that WW and B do not compete for resources, and the mutual benefits of proximity between the two species promote biomass formation.

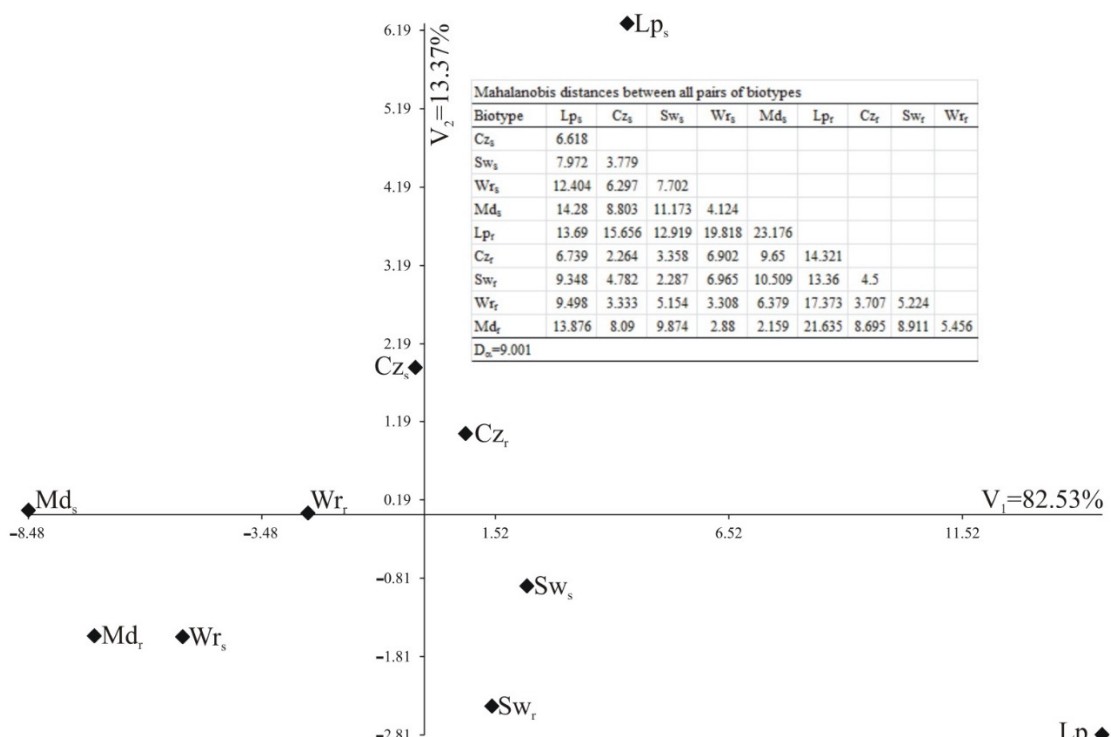

The table within the figure reads:

| Mahalanobis distances between all pairs of biotypes | | | | | | | | | |
|---|---|---|---|---|---|---|---|---|---|
| Biotype | $Lp_s$ | $Cz_s$ | $Sw_s$ | $Wr_s$ | $Md_s$ | $Lp_r$ | $Cz_r$ | $Sw_r$ | $Wr_r$ |
| $Cz_s$ | 6.618 | | | | | | | | |
| $Sw_s$ | 7.972 | 3.779 | | | | | | | |
| $Wr_s$ | 12.404 | 6.297 | 7.702 | | | | | | |
| $Md_s$ | 14.28 | 8.803 | 11.173 | 4.124 | | | | | |
| $Lp_r$ | 13.69 | 15.656 | 12.919 | 19.818 | 23.176 | | | | |
| $Cz_r$ | 6.739 | 2.264 | 3.358 | 6.902 | 9.65 | 14.321 | | | |
| $Sw_r$ | 9.348 | 4.782 | 2.287 | 6.965 | 10.509 | 13.36 | 4.5 | | |
| $Wr_r$ | 9.498 | 3.333 | 5.154 | 3.308 | 6.379 | 17.373 | 3.707 | 5.224 | |
| $Md_r$ | 13.876 | 8.09 | 9.874 | 2.88 | 2.159 | 21.635 | 8.695 | 8.911 | 5.456 |
| $D_\alpha = 9.001$ | | | | | | | | | |

**Figure 11.** Distribution of combinations of sites and herbicide-resistance (R)/susceptibility (S), based on parameters of winter wheat (WW), in the space of the first two canonical variables ($V_1$ and $V_2$), in 2018–2019. Sites: Cz—Czesławice, Md—Mydlniki, Lp—Lipnik, Sw—Swojczyce, Wr—Wrocław. Subscript "r" denotes the competition of WW with R and the subscript "s" with S.

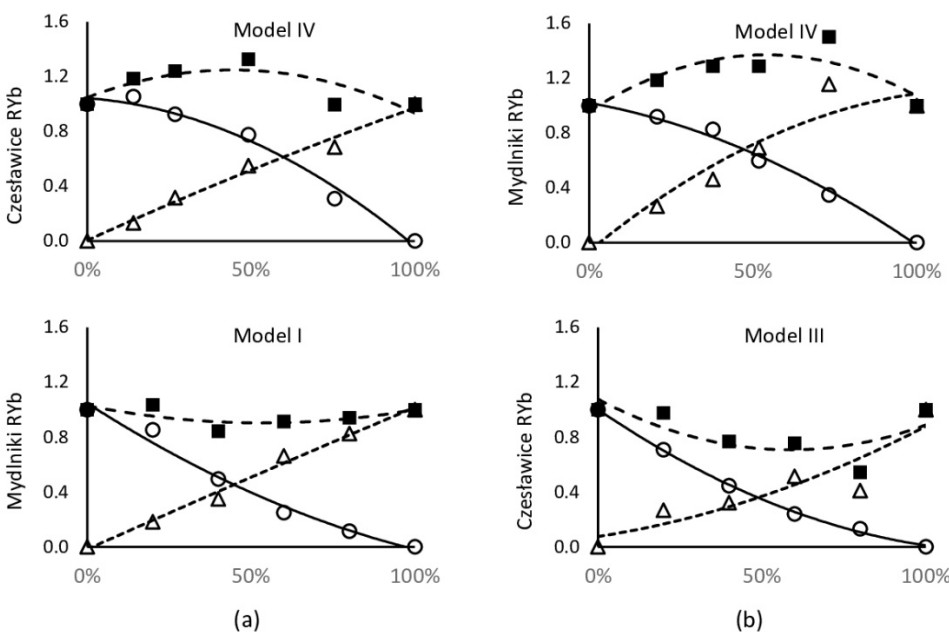

**Figure 12.** The replacement competition model for the relative biomass yield (RYb) of winter wheat and cornflower (B) in the 2019–2020 season, classified according to the *t*-test and *p*-values < 0.05. (**a**) competition between WW and B resistant (R) and (**b**) competition between WW and herbicide-susceptible cornflower (S). Legend: o—WW; Δ—R (**a**) or S (**b**), □—WW + R (**a**) or WW + S (**b**).

In Mydlniki, the model of competition between cornflower and wheat in forming the relative number of seeds (Figure 13, Table S13) was also the same for biomass formation. In

Czesławice, on the other hand, for both biotypes (R and S), there was no competition in seed production between R or S and WW (model I).

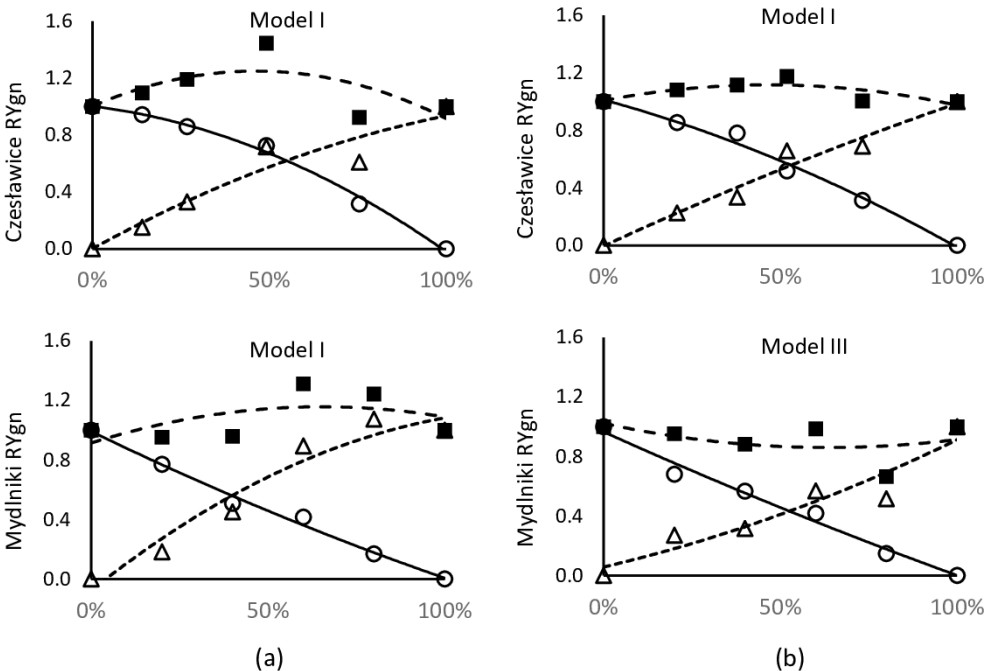

**Figure 13.** The replacement competition model for the relative seed number (RYse) of winter wheat and cornflower (B) in the 2019–2020 season, classified according to the *t*-test and *p*-values < 0.05. (**a**) competition between WW and B resistant (R) and (**b**) competition between WW and herbicide-susceptible cornflower (S). Legend: o—WW; Δ—R (**a**) or S (**b**), □—WW + R (**a**) or WW + S (**b**).

In the last growing season (2019/2020), there was no statistically significant effect of site, ratio, and biotype or their interaction on the competitive indices for relative seed number (CRs) and plant biomass (CRb) (Table S14).

In the last year of the study (2019–2020), CRs were not correlated with CRb ($r = 0.168$, $p = 0.691$) (Figure 14). Interestingly, of all the years of the study, CR rates had the lowest values. The greatest competitiveness of cornflower against wheat in terms of CRb and CRse was found in Mydlniki for the R biotype and WW 0.4 mixture (CRb = 0.6, CRse = 0.7). At the same location and WW share in the mixture of 0.4, a large competitive effect in biomass was also recorded for the S biotype (CRb = 0.8). In contrast, in terms of the number of seeds, resistant (R) cornflower was more competitive than susceptible (S) cornflower in Mydlniki. This relation was observed in each of the WW and B mixtures tested. In Czesławice, cornflower with the S biotype was more competitive with wheat than weed with the R biotype (in terms of biomass).

Figure 15 shows a correlation coefficients matrix for the traits observed in the 2018–2020 season. Positive correlations were observed between followed pairs of traits: length-yield ($r = 0.659$), length-biomass ($r = 0.716$), length-K ($r = 0.728$), yield-TGW ($r = 0.704$), yield-biomass ($r = 0.832$), yield-seeds ($r = 0.911$), yield-K ($r = 0.732$), biomass-seeds ($r = 0.892$), biomass-K ($r = 0.953$) and seeds-K ($r = 0.771$). However, negative for: sand-length ($r = -0.728$), sand-yield ($r = -0.732$), sand-biomass ($r = -0.953$) and sand-seeds ($r = -0.771$).

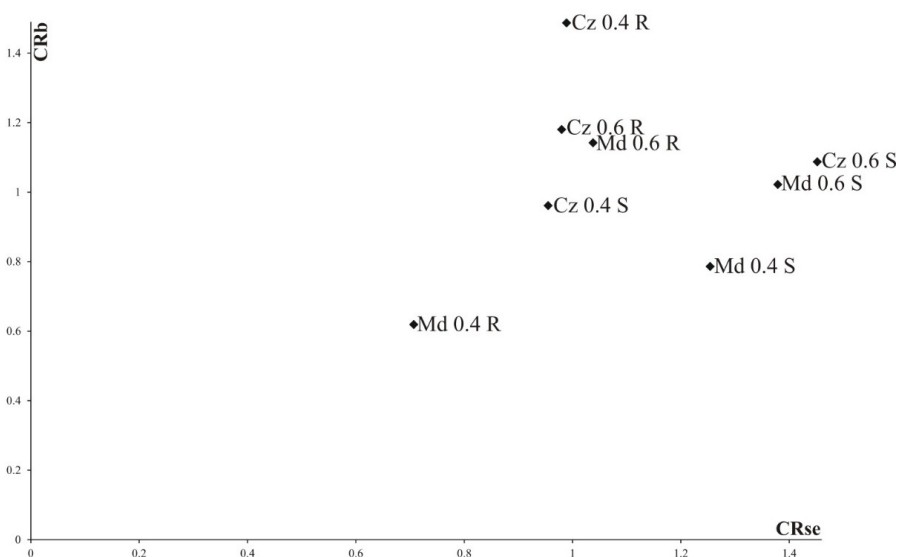

**Figure 14.** Relationships between relative biomass of plants (CRb) and the relative number of seeds (CRse) for the combinations of sites × winter wheat proportion in the mixture × cornflower biotypes, in 2019–2020. Legend: Cz—Czesławice, Md—Mydlniki (sites of experiments); R—cornflower with resistant biotype; S—susceptible biotype.

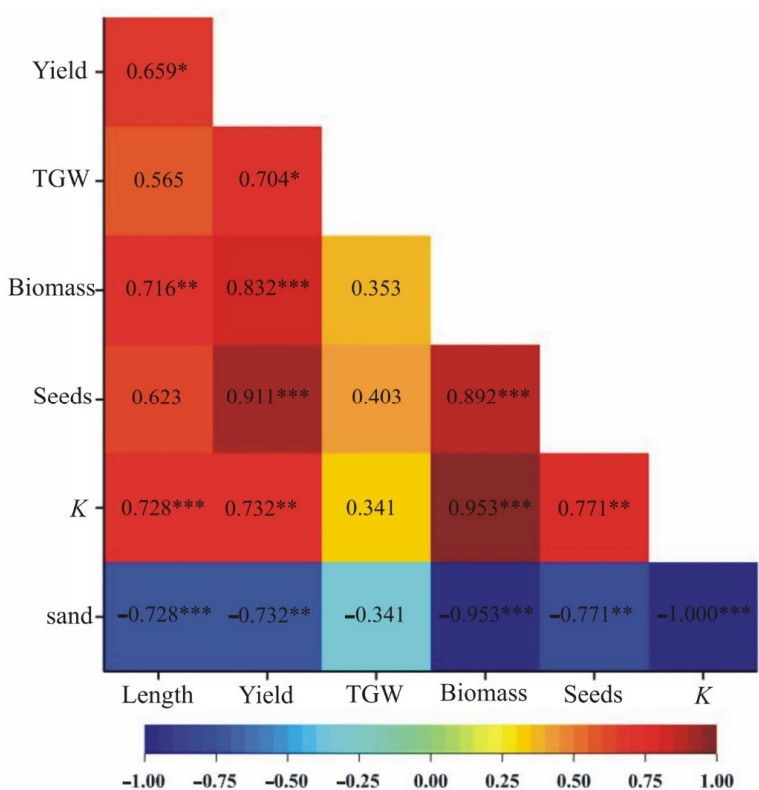

**Figure 15.** Heatmaps matrix for relationships between observed traits constructed on the basis of linear Pearson's correlation coefficients, in 2019–2020 season. *, **, ***— significant at level 0.05, 0.01, 0.001, respectively.

CVA performed in 2019–2020 season showed that the $V_1$ and $V_2$ accounted for 99.85% of the total multi-variability between combinations of studied factors (Figure 16). The $V_1$ was significantly negatively correlated with the plant's length, yield, plant biomass, number of seeds per plant, and days from sowing till emergence (Table S15). In the third year of the study, WW was found to compete best with cornflower in Mydlniki (optimal

season and silt loam soil), especially with its R biotype. In contrast, the weakest competitive ability of WW was found in Czesławice (humid season and silt loam soil). Interestingly, the resistant S biotype was less competitive against WW than the R biotype. The largest Mahalanobis distances (3.959) estimated on the basis of all seven traits jointly was found for $Md_s$ and $Cz_s$. The smallest Mahalanobis distances (0.861) was found between $Cz_r$ and $Cz_s$ (Figure 16).

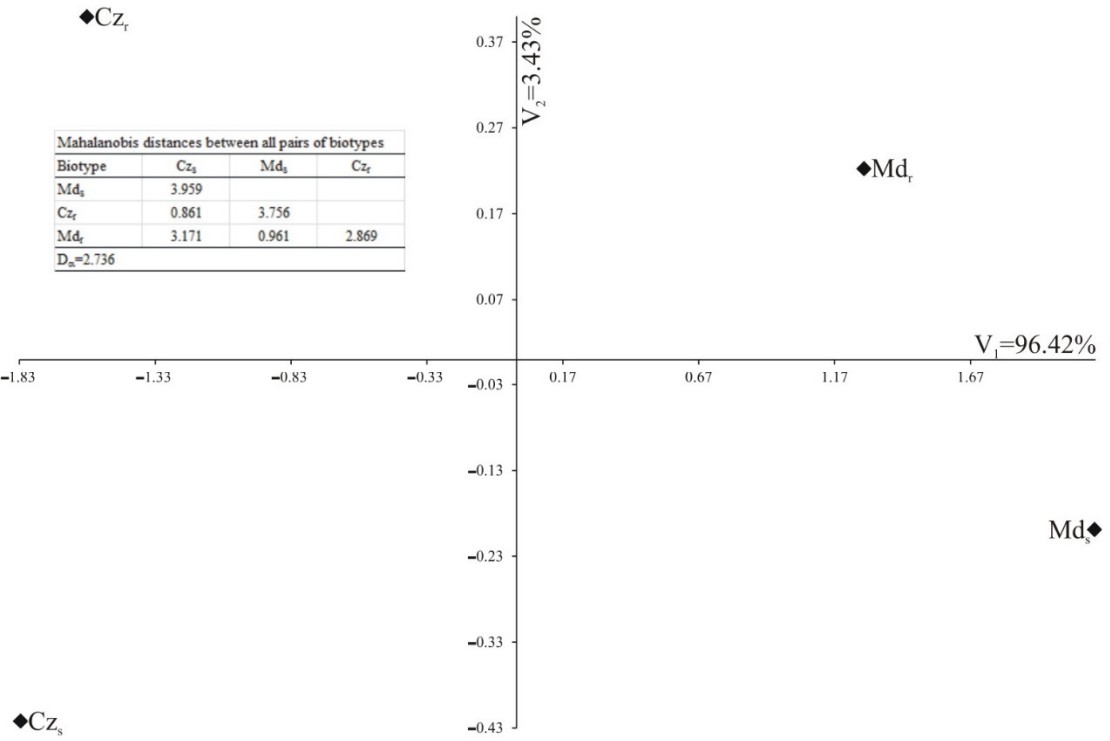

**Figure 16.** Distribution of combinations of sites and herbicide-resistance (R)/susceptibility (S), based on parameters of winter wheat (WW), in the space of the first two canonical variables ($V_1$ and $V_2$), in 2019–2020. Sites: Cz—Czesławice, and Md—Mydlniki. Subscript "r" denotes the competition of WW with R and the subscript "s" with S.

## 4. Discussion

During three growing seasons and at six locations, we conducted the experiments in situ as the replacement series model to study the competitive ability between winter wheat (WW) and cornflower resistant to florasulam and tribenuron-methyl (BR) as well as winter wheat and susceptible cornflower. On the basis of two indices, i.e., plant biomass and the number of seeds produced by tested plants, competitive relations were set out. Research sites were located on varying soils: from the lightest loamy sands (Lipnik), sandy loams (Winna Góra, Swojczyce), medium-heavy sandy clay loam (Wrocław), to heavy and compacted silt loams (Mydlniki, Czesławice). The years of studies showed differential weather: the first season (2017/2018) was relatively dry, the second one (2018/2019) generally dry, whereas the last one (2019/2020) was optimal or humid.

Under the relatively dry season, winter wheat (WW) generally showed great competitive ability with cornflower (B) at forming the relative number of seeds. Still, no significant competitive relationship between the tested species was found or found to be mutually antagonistic regarding biomass accumulation. In the dry year of the research, only three out of ten cases revealed wheat to be dominant in forming the number of seeds. At the same time, for none of them, there was an overall competitive effect of cornflower (R or S) against the tested cereal. Similar models were observed for the effects of competition between WW and B expressed in terms of relative biomass yield. In the wet season, on heavy soil (Czesławice), wheat and cornflower (R and S) did not compete for resources. The

mutual benefits of proximity between the two species favored biomass formation. In seed production, on the other hand, the ability of one species to disturb the other was equivalent.

To the best of our knowledge, no studies have been described in the scientific literature on the competitive ability of wheat against herbicide-resistant or herbicide-susceptible cornflower biotypes. However, the research on the competitiveness between wheat and herbicide-susceptible or resistant *Apera spica-venti* (silky bentgrass) [9] and *Alopecurus myosuroides* Huds. (blackgrass) [43] are available. During the wet season, wheat was more competitive with silky bentgrass or blackgrass biotypes, while under the dry season, the herbicide-resistant silky bentgrass biotype was more competitive in biomass accumulation. In contrast, in another experiment [51], *Raphanus raphanistrum* biotypes resistant and sensitive to ALS inhibitors were found to have greater competitive ability against wheat.

In general, water stress affects the competitiveness of wheat (WW) against cornflower (B) in both biomass accumulation and seed production. During dry years, WW's competitive ratio indices against B were higher in seed number (CRse) than in biomass (CRb). In the first season of the research, only in one case out of a total of 12 cases (Winna Gora, biotype S, WW share 0.4) cornflower was more competitive than wheat (CRb < 1; CRse < 1). In the next year of the research (dry season), the competition of WW against B for resources was lower. It depended more on the site than the cornflower biotype or the proportion of plants in the mixture. The lowest CRb indices were recorded in Wrocław (sandy clay loam soil) and the highest in Swojczyce (sandy loam). Under high or optimal precipitation (the third year of the study), the competitiveness of WW against B was significantly lower than in years with rainfall deficit. According to Guillemin et al. [23], the course of the weather, especially in spring and summer, provokes different competitive effects between wheat and cornflower. The authors demonstrated that under field conditions with less favorable growth for the wheat (lower availability of water in late spring), the occurence of *Centaurea cyanus* L. at densities of around 10–50 plants m$^{-2}$ was detrimental to the wheat. However, under conditions suitable for wheat growth, the company of cornflower could thus come up with stabilizing the yield of wheat.. Guillemin et al. [23] explain these relations by the positive role of the cornflower as a habitat for insects that are natural weeds pests. Epperlein et al. [52] also suggest that cornflowers could provide services for the protection of wheat. In other studies [53], just as in our research, summer drought promoted the competitiveness of wheat over field poppy and field pansy. On the contrary, wheat has been shown to suppress these weeds, especially under wet years, for competition between WW and blackgrass [43] or silky bentgrass [9].

Canonical variate analysis (CVA) clarified WW's competition against R and S biotypes at differentiated sites in Poland. In the first rather dry season, wheat at the Winna Góra site (central Poland, medium soils) was characterized by the best traits in competition with both cornflower biotypes. But in Wrocław (southwestern Poland, with medium-heavy soils) resistant biotype was less competitive with WW than the sensitive one. Under significant water shortage (the second growing season), on light soils (Lipniki, northwest), wheat was more competitive against the herbicide-susceptible (S) biotype.On heavy soils (Czesławice, southeast), greater WW competitive ability was revealed against the herbicide-resistant cornflower (R). Under optimal moisture (the last year of the experiment) on heavy soil (Mydlniki, southern Poland), wheat competed better with the R biotype than with the S biotype. WW's competition with the resistant or susceptible cornflower biotype was largely determined by soil type and weather conditions, indicating the high plasticity of the weed. The relevant factor affecting wheat's competitive ability against cornflower is weed seedlings' emergence time. In our study, cornflowers generally appeared a few days after wheat emergence or less frequently, along with wheat. This relationship has been confirmed in several other studies [54–57], which have shown that the emergence of wheat before weeds promote cereal competition against weedy plants.

## 5. Conclusions

Reffering to the replacement series competition model, it came out that weather conditions significantly affected the competitive ability of WW against R or S cornflowers (*Centaurea cyanus* L.). Water stress generally promoted increased competitiveness of wheat (WW) against cornflower (B) in both biomass accumulation and seed production. In addition, wheat competition against cornflowers may have been influenced by the 12–0 days earlier emergence of wheat than that of cornflowers. There was no clear effect of soil type (location) and cornflower S or R biotype on the tested indices. However, it was noticed that wheat was more competitive against the herbicide-susceptible (S) biotype on light soils. In contrast, greater WW competitive ability was revealed against herbicide-resistant (R) cornflower on heavy soils.

In conclusion, the competitiveness of winter wheat against herbicide-resistant or herbicide-sensitive cornflower biotypes is significantly dependent on habitat conditions. It is, therefore, reasonable to study this phenomenon in more detail. It would also be interesting to learn about the underground competition on varying soil types and under variable water availability. Research at the molecular level, leading to understanding the mechanism and improving crop productivity, is also recommended.

**Supplementary Materials:** The following supporting information can be downloaded at: https://www.mdpi.com/article/10.3390/agronomy12112751/s1, Table S1: Sum of precipitation and mean temperatures during season 2017/18 in the sites of study. Table S2: Sum of precipitation and mean temperatures during season 2018/19 in the sites of study. Table S3: Sum of precipitation and mean temperatures during season 2019/20 in the sites of study. Table S4: The *t*-test values and *p*-values for comparison between empirical and theoretical models for biomass competition in replacement series design in 2017–2018. Table S5: The *t*-test values and *p*-values for comparison between empirical and theoretical models for number of seeds competition in replacement series design in 2017–2018. Table S6: Mean squares from the three-way analysis of variance for the competitive ratio of winter wheat (WW) and herbicide-resistant or susceptible cornflower (B) calculated for the relative plants' biomass (CRb) and relative seed number (CRse) at two plant ratios 6WW:4B and 4WW:6B, in the seasons 2017–2018. Table S7: Results of discrimination analysis for the first and second canonical variables (CV1, CV2) for seven parameters of winter wheat (WW) in competition with herbicide-resistant or susceptible cornflower (B) in the season 2017–2018, depending on hydrothermal conditions and soil texture at the study sites. Table S8: The *t*-test values and *p*-values for comparison between empirical and theoretical models for biomass competition in replacement series design in 2018–2019. Table S9: The *t*-test values and *p*-values for comparison between empirical and theoretical models for number of seeds competition in replacement series design in 2018–2019. Table S10: Mean squares from the three-way analysis of variance for the competitive ratio of winter wheat (WW) and herbicide-resistant or susceptible cornflower (B) calculated for the relative plants' biomass (CRb) and relative seed number (CRse) at two plant ratios 6WW:4B and 4WW:6B, in the seasons 2018–2019. Table S11: Results of discrimination analysis for the first and second canonical variables (CV1, CV2) for seven parameters of winter wheat (WW) in competition with herbicide-resistant or susceptible cornflower (B) in the season 2018–2019, depending on hydrothermal conditions and soil texture at the study sites. Table S12: The *t*-test values and *p*-values for comparison between empirical and theoretical models for biomass competition in replacement series design in 2019–2020. Table S13: The *t*-test values and *p*-values for comparison between empirical and theoretical models for number of seeds competition in replacement series design in 2019–2020. Table S14: Mean squares from the three-way analysis of variance for the competitive ratio of winter wheat (WW) and herbicide-resistant or susceptible cornflower (B) calculated for the relative plants' biomass (CRb) and relative seed number (CRse) at two plant ratios 6WW:4B and 4WW:6B, in the seasons 2019–2020. Table S15: Results of discrimination analysis for the first and second canonical variables (CV1, CV2) for seven parameters of winter wheat (WW) in competition with herbicide-resistant or susceptible cornflower (B) in the season 2019–2020, depending on hydrothermal conditions and soil texture at the study sites.

**Author Contributions:** Conceptualization, A.S., R.W., E.T. and K.M.; methodology, A.S., J.B. and R.W.; investigation, R.W., E.T., A.S., C.P., K.D., K.M., E.K.-P. and M.P.; resources, R.W., E.T., A.S., C.P., K.D., K.M., E.K.-P. and M.P.; data curation, R.W, E.T., A.S. and J.B.; writing—original draft

preparation, R.W., E.T., A.S. and J.B.; writing—review and editing, R.W., E.T., A.S., K.M., J.B. and M.P.; visualization, R.W., A.S. and J.B.; supervision, A.S. and K.M.; funding acquisition, A.S. and K.M. All authors have read and agreed to the published version of the manuscript.

**Funding:** This research was funded by The National Centre for Research and Development, contract number: BIOSTRATEG 3/347445/1/NCBR/2017.

**Data Availability Statement:** The data presented in this study are available on request from the corresponding author.

**Conflicts of Interest:** The authors declare no conflict of interest.

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
