# Peer review of "Competition between Winter Wheat and Cornflower (Centaurea cyanus L.) Resistant or Susceptible to Herbicides under Varying Environmental Conditions in Poland"

_agronomy, doi:10.3390/agronomy12112751_

Round 1

Reviewer 1 Report

This article examined Competition Between Winter Wheat and Cornflower (Centaurea cyanus L.) Resistant or Susceptible to Herbicides Under Varying Environmental Conditions in Poland. It will help the to understand stress or environmental management of the crops and applications of herbicides. Before recommending this article for publication, there are some shortcomings for that should be resolve.

General comments

Assign line numbers.

Abstract

In abstract section results are not clear. In addition, results are presented in very general way. Specific or quantitative results should be provided.

Also add conclusion and future perspective in the abstracts  

Introduction

First line of para 4 is repetition.

Add medicinal and economic importance of the corn flower.

Origin of the corn flower should be provided.

Paragraph 2 line “by environmental factors such as weather and soil conditions” should be cited with relevant study. https://doi.org/10.3390/ijms22179175,

Materials and Methods

Experiment is well designed, and methodology is well written, But in some places language must be revised and citations are required for methodologies.

Results and Discussion

The result and discussion are well presented but grammatical mistakes must be revised by the authors.

Conclusion is well justified. The authors should discuss some points for the future studies molecular level studies are required to know about the involved mechanism and improvement of productivity in crops.

Author Response

Respond to the Reviewers' comments on the article: ”Competition Between Winter Wheat and Cornflower (Centaurea cyanus L.) Resistant or Susceptible to Herbicides Under Varying Environmental Conditions in Poland”    

by Roman Wacławowicz , Ewa Tendziagolska, Agnieszka Synowiec, Jan Bocianowski, Cezary Podsiadło, Krzysztof Domaradzki, Katarzyna Marcinkowska, Ewa Kwiecińska-Poppe, Mariusz Piekarczyk

The authors would like to thank for sending reviews of the paper. We have responded positively to all the comments and they have been included in the new version of the article.

Review no. 1

  • In section ”Abstract” a description of the research results was detailed. Information on future research has also been added.
  • In section: „Introduction”, according to Reviewer suggestion, excerpts of previously repeated content have been removed.
  • The medicinal and economic importance of cornflower and its origin were added.
  • The introduction was supplemented by the reviewer's suggested publication.
  • In the "Material and methods" section, literature on the analyses performed was added.
  • The conclusions have been supplemented by more detailed research and other areas of research related to the presented issue have been also indicated.
  • English language has been improved.

Reviewer 2 Report

The work “Competition Between Winter Wheat and Cornflower (Centaurea cyanus L.) Resistant or Susceptible to Herbicides Under Varying Environmental Conditions in Poland” seems to have been well conducted and well analyzed and it is well written.

I suggest some minor changes:

Do not repeat title words in the keywords.

Abstract: “Competitive ability of cereals against segetal weeds depends on many factors.” Please avoid terms like “many factors”, specify the information better.

Page 2 Introduction: “The presence of weeds in crop fields is undesirable for several reasons.” Please avoid terms like “several reasons”, specify the information better.

Page 2: “Of the four most commonly presented models (additive, substitutive, systematic, and neighborhood) proposed by Radosevich [32], the most suitable for our experiment is the substitutive series model [37], where the total plant density is kept constant….. Based on a replacement series model,…”

Are you using “replacement series model” as a synonym for “substitutive series model”? I suggest you standardize the term throughout the work and also in the legend of the figures.

Page 5: replace “1000-rain” with “1000-grain”

Figures 2 and 3, 7 and 8, 12 and 13: the Y axis can be in black color, and please check if it would be a comma or a period.

Figures 6, 11, 16: In the excel table inserted in the figure, I suggest keeping the entire background white, so that the gray lines do not appear. And check if it is possible to increase the resolution of this part.

Tables S1, S2, and S3: I suggest reversing the direction of texts Winna Góra Swojczyce WrocÅ‚aw Winna Góra Swojczyce WrocÅ‚aw,….

Author Response

Respond to the Reviewers' comments on the article: ”Competition Between Winter Wheat and Cornflower (Centaurea cyanus L.) Resistant or Susceptible to Herbicides Under Varying Environmental Conditions in Poland”   

by Roman Wacławowicz , Ewa Tendziagolska, Agnieszka Synowiec, Jan Bocianowski, Cezary Podsiadło, Krzysztof Domaradzki, Katarzyna Marcinkowska, Ewa Kwiecińska-Poppe, Mariusz Piekarczyk

The authors would like to thank for sending reviews of the paper. We have responded positively to all the comments and they have been included in the new version of the article.

Review no. 2

  • In accordance with the reviewer's recommendation, the keywords were changed.
  • Indicated issues from the abstract and introduction have been corrected.
  • The name for the replacement series model has been standardized (instead of „substitutive” into „replacement”).
  • Figures 2,3,7,8,12,13 have been revised (colour has been changed and commas replaced with periods) as well as figures 6, 11,16.
  • In suplement in tables S1, S2, S3 we did not change the direction of the text, as this would have increased the size of the table and thus resulted in poorer readability of the test results.

Reviewer 3 Report

Very interesting material. One comment BBCH stage winter wheat 99 is harvested product, not mature plant.

Author Response

Respond to the Reviewers' comments on the article: ”Competition Between Winter Wheat and Cornflower (Centaurea cyanus L.) Resistant or Susceptible to Herbicides Under Varying Environmental Conditions in Poland"                                    

by Roman Wacławowicz , Ewa Tendziagolska, Agnieszka Synowiec, Jan Bocianowski, Cezary Podsiadło, Krzysztof Domaradzki, Katarzyna Marcinkowska, Ewa Kwiecińska-Poppe, Mariusz Piekarczyk

The authors would like to thank for sending reviews of the paper. We have responded positively to all the comments and they have been included in the new version of the article.

Review no.3

According to Reviewer’s suggestion development stage has been corrected (BBCH 89 instead of 97-99).